

# Clinical, serological and epidemiological features of hepatitis A in León, Nicaragua

Sophie Jaisli[1], Orlando Mayorga[2], Nadia Flores[2], Sandra de Berti[3], Gustav Frösner[4], Christian Herzog[1,5,6], Marcel Zwahlen[1] and Sereina A. Herzog[7,8]

[1] Institute of Social and Preventive Medicine, University of Bern, Bern, Switzerland
[2] Department of Microbiology & Parasitology, Faculty of Medical Sciences, National Autonomous University, León, Nicaragua
[3] Casa B5, Lomas de Monserrat, Managua, Nicaragua
[4] Institute of Virology, Technical University, Munich, Germany
[5] Medical Department, Swiss Tropical and Public Health Institute, Basel, Switzerland
[6] University of Basel, Basel, Switzerland
[7] Centre for Health Economics Research and Modelling of Infectious Diseases, Vaccine & Infectious Disease Institute, University of Antwerp, Wilrijk, Belgium
[8] Institute for Medical Informatics, Statistics and Documentation, Medical University of Graz, Graz, Austria

Corresponding author
Christian Herzog,
herzog.ch47@gmail.com

## ABSTRACT

**Background and Objectives**. To monitor and document the endemicity and disease burden of acute hepatitis A in the area of an ongoing vaccine effectiveness study in León, Nicaragua.

**Methods**. At community health centres in León, all children, adolescents and young adults presenting with jaundice and/or other clinical signs of hepatitis were offered free serologic screening (hepatitis A, B and C) and blood tests for liver enzymes and bilirubin. Clinical and socioeconomic data were collected with a structured questionnaire. Diagnosis of acute hepatitis A was confirmed by anti-HAV IgM testing. Using logistic regression we compared the characteristics and living conditions of acute hepatitis A cases with those of non-cases.

**Results**. Of 557 eligible subjects enrolled between May 2006 and March 2010, 315 (56.6%) were diagnosed with hepatitis A, 80.6% of them ≤10 years and five >18 years of age. No severe cases were encountered. Apart from jaundice (95.6%) and other signs of hepatitis A (fever, pale stool, dark urine, nausea, vomiting, anorexia), two thirds of patients had moderately raised liver enzymes. Cases occurred throughout the year, with highest incidences from August to March. Poor sanitary conditions and crowding were the main risk factors.

**Conclusions**. In the study area, hepatitis A is still highly endemic in young and school age children living in low socioeconomic conditions. There are, however, first indications that the endemicity level is shifting from high to high-intermediate.

## INTRODUCTION

The hepatitis A virus (HAV) is mainly transmitted by the fecal-oral route (*World Health Organization, 2010a*). Therefore, the burden of disease correlates with the quality of sanitary conditions, the access to clean water and the socioeconomic status (*Jacobsen & Koopman, 2005*). After an incubation period of 14 to 50 days the disease might start with unspecific symptoms such as malaise, fatigue, anorexia, vomiting, abdominal discomfort and diarrhea. In the course of symptomatic illness, which ensues more often in older children and adults, dark urine, pale stool and jaundice can also be observed; in children below 6 years of age less than 10% show jaundice (*Armstrong & Bell, 2002*). Rarely, fulminant hepatic failure develops, associated with a severe disease process and a high case fatality, usually necessitating liver transplantation (*Lemon et al., 2018*). Interestingly, HAV has been reported to be a major cause for fulminant hepatic failure in children in Pakistan (*Talat et al., 2020*) and Latin America (*Ciocca et al., 2007*).

There are many unreported cases of hepatitis A (up to 80% of infections). WHO estimated from 1990 to 2005 an increase in the number of infections and deaths due to hepatitis A from 177 to 212 million and 30,283 to 35,245, respectively (*World Health Organization, 2010a*). From the 212 million infections only 33 million cases are estimated to present as symptomatic illness (*Rein et al., 2014*). Different studies showed over the last two decades, however, decreasing hepatitis A seroprevalences all over the world, except for Africa, most likely due to better access to clean water, better sanitary conditions, higher educational level and less crowded housing conditions (*Jacobsen & Wiersma, 2010*). Levels of hepatitis A endemicity are defined on the basis of seroprevalence: "high (≥ 90% by age 10 years); intermediate (≥ 50% by age 15 years, with <90% by age 10 years); low (≥ 50% by age 30 years, with <50% by age 15); and very low (<50% by age 30 years)" (*Jacobsen & Wiersma, 2010*). Because HAV infection leads to lifelong immunity there are in high endemic settings hardly any susceptible adults, and clinically manifest HAV infections are rare. However, if sanitary conditions improve, there will be a shift to intermediate endemicity, and with increasingly older age groups remaining susceptible, more severe cases (*World Health Organization, 2010a*) and outbreaks (*Bell et al., 1998*) will be observed. For Nicaragua there exist very few published data on the hepatitis A epidemiology (*Andani et al., 2020*). Hepatitis A infections are not routinely reported by the health authorities in Nicaragua, a still highly endemic country with 36.9 cases per 100,000 inhabitants reported for 2012 (Supplemental File 4).

Universal mass vaccination (UMV) of toddlers with a two-dose regimen is now recommended by WHO for countries with sizable acute hepatitis A incidences and declining endemicity (*World Health Organisation, 2012*). Two-dose UMV successfully eliminates hepatitis A within a few years, as shown, e.g., in Israel (*Levine et al., 2015*). Single-dose UMV likewise lowers rapidly the disease burden, as documented for Argentina (*Vizzotti et al., 2014*) and Brazil (*Souto, de Brito & Fontes, 2019*). Although single-dose vaccination elicits in >95% of children seroprotection for at least 7–10 years (*Mayorga et al., 2016*; *Urueña et al., 2016*; *Espul et al., 2020*), the long-term protective effectiveness over 20-30 years remains still to be established.

The objectives of our project were: (1) to document with a prospective survey (2006–2010)—in parallel to an ongoing single-dose vaccine effectiveness study (*Mayorga et al., 2016*)—the sustained circulation of HAV in León, Nicaragua; (2) to describe the clinical features of acute hepatitis A; and (3) to evaluate the HAV infection risk factors in the same study area, for which already for the 1990s and the 2000s a high endemicity had been reported (*Mayorga Perez et al., 2014*).

## MATERIALS & METHODS

### Study conduct

A viral hepatitis diagnosis survey was conducted from May 2006 to June 2010 to document—among others—the continuing circulation of HAV in the study area of a concurrent single-dose hepatitis A vaccine trial (*Mayorga et al., 2016*). The survey was set up by the Medical Faculty of the National Autonomous University (UNAN) in León to capture children, adolescents and adults with clinical hepatitis. During the study period regular advertisements were made in the local radio program. Everyone who presented symptoms suggestive for acute hepatitis, such as jaundice, fever, vomiting, nausea, etc. was encouraged to visit one of the community health centers in order to have his/her illness assessed for free. Reporting patients were asked if they would agree to participate in the study, which consisted—apart from answering a questionnaire—only in taking of a single blood sample (5–7 ml). These patients were thus offered a free serologic screening to establish the viral etiology (hepatitis A, B or C), as well as free blood tests for liver enzymes and bilirubin. The presence of diagnostic anti-HAV IgM antibodies was later confirmed at the Technical University, Munich, Germany. The study was a community-based survey and no specific inclusion or exclusion criteria were applied.

The project was approved by the Medical Faculty of the UNAN, León. Written informed consent was obtained from participating patients and in case of children/adolescents from their parents or guardians.

### Questionnaire

The questionnaire comprised questions on clinical features and demographic characteristics (age and sex), as well as on socioeconomic conditions like crowding (number of rooms and people per household), location of the main water source and the toilets, and some data regarding education and work. Symptoms (anorexia, nausea, vomiting, fever, malaise) and clinical signs (jaundice, pale stool, dark urine) were solicited and documented. The days from start of symptoms until presentation at the health centers and the history of a recent blood transfusion were also noted.

### Laboratory evaluation

Screening of serum samples for anti-HAV IgM (as for Hepatitis B and Hepatitis C) was done daily at the clinical microbiology laboratory of the Medical Faculty of UNAN León, using the enzyme immunoassay (EIA) stripe test kit ImmunoComb II from Orgenics, Israel. The sera were stored at −20 °C and later shipped to the Institute of Virology, Technical University, Munich, Germany, for confirmative analysis of anti-HAV IgM antibodies

(*CDC, 2020*), using the microparticle EIA HAVAB-M 2.0 for the AxSYM system, Abbott Laboratories, USA. The following Index reference values (anti-HAV-IgM content) were used: <0.80: non-reactive test, acute hepatitis A excluded; 0.80–1.20: gray-zone, possibly recent recovery from acute HAV infection; >1.20: reactive, indicating acute HAV infection (Frösner G., personal communication). "Gray-zone" results were counted as no hepatitis A.

Quantitative analysis of blood samples for total anti-HAV (IgG+IgM) antibodies were made at the same laboratory, using the microparticle EIA HAVAB 2.0 Quantitative for AxSYM, Abbott Laboratories, USA. A result of $\geq 20$ mIU/mL was defined as seroprotective (*CDC, 2020*), with a lower limit of detection of 10 mIU/mL.

Reference normal values for GPT were $\leq 40$ U/l for men and $\leq 32$ U/l for women using the GPT (ALT)-LQ test. For GOT they were $\leq 38$ U/l for men and $\leq 31$ U/l for women, using the GOT (AST)-LQ test. Total bilirubin values were considered normal up to 1.10 mg/dl, using the BILIRUBIN T-DMSO test. All tests were from SPINREACT, S.A./S.A.U, Spain.

## Statistical methods

For the descriptive analysis of categorical characteristics of acute hepatitis A cases and non-cases we calculated the chi-square statistics and *p*-values for the null hypothesis of difference in distribution of categories. Univariable and multivariable logistic regression analysis was used to assess associations of possible risk factors for being a hepatitis A case, such as excreta disposal, water source (yes vs. no), crowding (>2.5 persons per room in a given household vs $\leq 2.5$), age groups and sex. From the logistic regressions we report Odds Ratios and 95% confidence intervals (CIs). We also calculated the geometric mean concentrations (GMCs) with 95% CIs for the total anti-HAV antibodies. For values <10 mIU/ml, a value of 5 mIU/ml was used for the GMC calculation. A *p*-value <0.05 was considered as significant. Statistical Software R (4.0.3) was used for the statistical analyses.

# RESULTS

## Study population

A total of 575 children, adolescents and adults with suspected viral hepatitis were screened from May 2006 until June 2010. Eighteen patients had to be excluded from analysis: samples lost ($n = 2$), anti-HAV IgM serology for April-June 2010 samples lacking ($n = 15$) or unclear age ($n = 1$), thus leaving 557 eligible patients (see Study Flow Chart, Fig. 1). Six patients had "gray-zone" results of the anti-HAV IgM test and were counted as non-hepatitis A. The majority (80.6%) of the 315 patients with acute hepatitis A were children $\leq 10$ years of age, two were $\leq 1$ year and five >18 years old (Table 1). With this cohort of acute hepatitis A cases we documented the continuous exposure to HAV of the subjects enrolled into the single-dose hepatitis A vaccination trial carried out by our team in the same area during 2005–2012 (*Mayorga et al., 2016*).

We did not evaluate occupation and educational levels because the vast majority of study subjects were very young or school age children and thus an analysis of these parameters would not have made sense.
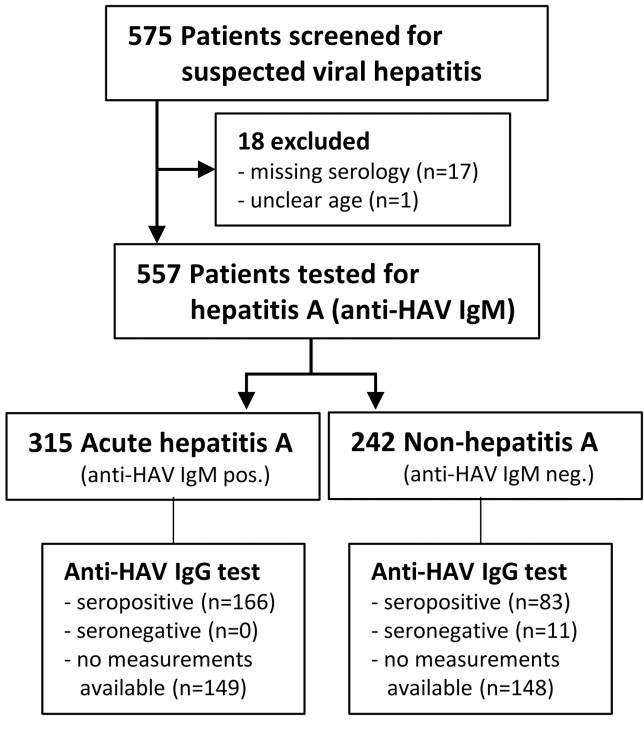

**Figure 1** Study flowchart.

## Seroprevalence and antibody levels

Quantitative anti-HAV IgG+IgM measurements could only be done in 166 of the 315 hepatitis A cases (data not shown) and in 94 of the 242 cases without hepatitis A (Table 2). All patients with acute hepatitis A had high anti-HAV antibody concentrations, with minimum and maximum levels of 1,485 mIU/ml and 181,000 mIU/ml, respectively. Eighty-three (88.3%) of the 94 non-hepatitis A patients had protective levels of anti-HAV antibodies (min 62 mIU/ml –max 209,900 mIU/ml), representing past HAV infection. Three of the 5 children aged ≤1 year had protective levels of maternal anti-HAV antibodies of 258, 341 and 6,173 mIU/ml, respectively, and 2 were anti-HAV negative. The non-hepatitis A cases aged 1-≤6 years, 6-≤11 years and the >11 years old subjects had had hepatitis A in the past in 68.8%, 89.5% and 96.3%, respectively. Among the subjects >11 years of age only two young adults aged 18 and 19 years were anti-HAV antibody negative. Only 3 and 8 of the 557 patients were screened positive (IgM antibodies) for acute hepatitis B and acute hepatitis C, respectively. These cases were not further investigated within this project.

## Clinical features

For 15 patients the start of symptoms was unclear or missing. 56.8% of the remaining 542 patients presented themselves within 5 days after falling ill and one quarter after 6 to 11 days. Among the 89 patients presenting later than 11 days there were proportionally less cases of hepatitis A, i.e., 37.1% as compared to 62.9%. The leading hepatitis A feature

**Table 1  Characteristics of the study population.**

| Characteristic | Category | N (%) | | p-value |
|---|---|---|---|---|
| | | **Acute Hepatitis A** (*n* = 315) | **Non Hepatitis A** (*n* = 242) | |
| *Demographics* | | | | |
| Age | | | | <0.001 |
| | ≤1 year | 2 (0.6) | 16 (6.6) | |
| | 1–≤5 years | 70 (22.2) | 47 (19.4) | |
| | 5–≤10 years | 182 (57.8) | 64 (26.4) | |
| | 10–≤14 years | 46 (14.6) | 32 (13.2) | |
| | 14–≤18 years | 10 (3.2) | 11 (4.5) | |
| | >18 years | 5 (1.6) | 72 (29.8) | |
| Sex | | | | 1.000 |
| | female | 157 (49.8) | 120 (49.6) | |
| | male | 158 (50.2) | 122 (50.4) | |
| *Socioeconomic* | | | | |
| Housing | | | | 0.013 |
| | no crowding | 141 (44.8) | 135 (55.8) | |
| | crowding | 174 (55.2) | 107 (44.2) | |
| Toilets | | | | 1.000 |
| | outside | 157 (49.8) | 99 (40.9) | |
| | inside | 158 (50.2) | 143 (59.1) | |
| Water | | | | 0.044 |
| | outside | 57 (18.1) | 43 (17.8) | |
| | inside | 258 (81.9) | 199 (82.2) | |
| *Clinical features* | | | | |
| Jaundice | | | | <0.001 |
| | no | 14 (4.4) | 48 (19.8) | |
| | yes | 301 (95.6) | 194 (80.2) | |
| Pale stool | | | | <0.001 |
| | no | 207 (65.7) | 203 (83.9) | |
| | yes | 108 (34.3) | 39 (16.1) | |
| Dark urine | | | | <0.001 |
| | no | 120 (38.1) | 162 (66.9) | |
| | yes | 195 (61.9) | 80 (33.1) | |
| Nausea | | | | <0.001 |
| | no | 83 (26.3) | 98 (40.5) | |
| | yes | 232 (73.7) | 144 (59.5) | |
| Vomiting | | | | 0.001 |
| | no | 81 (25.7) | 127 (52.5) | |
| | yes | 234 (74.3) | 115 (47.5) | |

**Table 1** (*continued*)

| Characteristic | Category | N (%) | | p-value |
|---|---|---|---|---|
| | | **Acute Hepatitis A** ($n = 315$) | **Non Hepatitis A** ($n = 242$) | |
| Fever | | | | 0.014 |
| | no | 64 (20.3) | 72 (29.8) | |
| | yes | 251 (79.7) | 170 (70.2) | |
| Anorexic | | | | 0.004 |
| | no | 55 (17.5) | 68 (28.1) | |
| | yes | 260 (82.5) | 174 (71.9) | |
| Malaise | | | | 0.101 |
| | no | 46 (14.6) | 49 (20.2) | |
| | yes | 269 (85.4) | 193 (79.8) | |
| *Laboratory finding* [*] | | | | |
| Liver enzymes GOT | | | | <0.001 |
| | elevated | 281 (89.2) | 108 (44.6) | |
| | normal | 34 (10.8) | 134 (55.4) | |
| Liver enzymes GTP | | | | <0.001 |
| | elevated | 289 (91.7) | 93 (38.4) | |
| | normal | 26 (8.3) | 149 (61.6) | |
| Total serum bilirubin | | | | <0.001 |
| | elevated | 274 (87.0) | 118 (48.8) | |
| | normal | 41 (13.0) | 124 (51.2) | |

**Notes.**
[*]cut-off definitions are described in the Methods section.

**Table 2 Anti-HAV IgG antibody levels of subgroup of patients ($n = 94$) without hepatitis A.**

| Patients tested | | Anti-HAV IgG antibody concentrations (mIU/mL) | | Seroprotection [**] |
|---|---|---|---|---|
| **n = 94** | | **GMC[*] (95% CI)** | **Min/max** | **n (%)** |
| Age | | | | |
| ≤1 year | 5 | 106.3 (2.4; 4,719.9) | <10/6,173 | 3 (60.0) |
| 1–≤6 years | 16 | 965.6 (111.7; 8,345.2) | <10/209,900 | 11 (68.8) |
| 6–≤11 years | 19 | 8,267.6 (1,889.9; 36,167.4) | <10/147,100 | 17 (89.5) |
| 11–≤16 years | 13 | 10,568.9 (5,178.7; 21,569.4) | 989/50,380 | 13 (100.0) |
| 16–≤20 years | 7 | 1,004.6 (30.8; 32,751.5) | <10/32,300 | 5 (71.4) |
| 20–≤25 years | 9 | 11,990.0 (6,557.8; 21,921.8) | 3,529/29,050 | 9 (100.0) |
| >25 years | 25 | 5,487.8 (3,468.7; 8,682.1) | 602/34,150 | 25 (100.0) |

**Notes.**
GMC: geometric mean concentration; CI: confidence interval.
[*]For <10 mIU/ml a value of 5 mIU/ml was used for the GMC calculation.
[**]Seroprotection: ≥20 mIU/mL of anti-HAV IgG antibodies.

was jaundice. Of all 557 patients with assumed viral hepatitis 495 (88.9%) had jaundice, 301 (95.6%) among the acute hepatitis A patients and 194 (80.2%) among the non-A cases ($p < 0.001$) (Table 1). From the 62 cases without jaundice only 14 (22.6%) had acute hepatitis A. Hepatitis A patients had significantly ($p < 0.001$) more of the following

solicited symptoms and signs, than non-A patients: 34.3% vs. 16.1% pale stool, 61.9% vs. 33.1% dark urine, 73.7% vs. 59.5% nausea, and 74.3% vs. 47.5% vomiting. Rates of 79.7% vs. 70.2% were found for fever ($p = 0.014$), and 82.5% vs. 71.9% for anorexia ($p = 0.004$). Only the difference for malaise (85.4% vs. 79.8%) was not significant ($p = 0.101$) (Table 1). Having had a blood transfusion was more often recorded in subjects without hepatitis A (5.4% vs. 1.3%, $p = 0.011$), probably a chance finding due to the small numbers ($n = 17$).

## Laboratory findings (biochemistry)

Serum liver enzymes (GOT, GPT) and total serum bilirubin were all significantly ($p < 0.001$) more often elevated in hepatitis A than in non-A patients (Table 1). The highest values were 2920 U/l for GPT, 2750 U/l for GOT and 43.2 mg/dl for total bilirubin. According to the staff of the health centers none of the study patients developed a more severe pathology later on.

## Sociodemographic factors

The odd ratio (OR) for contracting hepatitis A was for children aged 5-≤10 years 2.49 (crude) and 2.45 (adjusted), respectively, and then declined with age to 0.06 (crude) and 0.06 (adjusted), respectively, in the patients aged >18 years (Table 3). Logistic regression did not change these values for the age groups, the $p$-value stayed significant with <0.001. Significantly less cases of acute hepatitis A were found in children from >10 years onwards. This fact was independent from other sociodemographic factors. The OR for having a toilet inside the house was 0.7 (crude) and 0.75 (adjusted), respectively, with $p$-values of 0.036 (crude) and 0.191 (adjusted), respectively, however, no longer significant after logistic regression. The place of the toilet was a factor probably influenced by other hygiene conditions and/or sex and age and was thus not an independent risk factor. OR for 'crowding' was 1.56 (crude) and 1.1 (adjusted), with $p$-values of 0.01 (crude) and 0.62 (adjusted), respectively; thus, also crowding was not found to be an independent risk factor (Table 3). Logistic regression analyses were also made for the combinations of age groups and crowding, age groups and water source, localization of toilet and sex, and again no significant findings were seen and no differences could be observed (data not shown).

## Seasonality

All along the observation period May 2006 to June 2010 cases of suspected viral hepatitis were enrolled (~11.1/month), with slightly more patients presenting themselves at the health centers in the 1st and 3rd quarters of the year (data not shown). 56.6% of suspected cases were diagnosed with acute hepatitis A (~6.7/month). On average, more cases of acute hepatitis A were diagnosed in the 4th and 1st quarters (autumn & winter) than in the 2nd and 3rd quarters (Fig. 2). The percentage of hepatitis A cases among all persons examined decreased from 63.5% ($n = 54/85$) in 2006 to 50.0% ($n = 32/64$) cases in 2010 ($p = 0.013$, p for trend = 0.0016). The age distribution of patients with acute hepatitis A remained, however, similar over the years (data not shown).

**Table 3  Sociodemographic factors, multiple logistic regression.**

| Characteristics | Total N | Hepatitis A patients n (% of N) | Crude OR (95% CI) | Adjusted OR[*] (95% CI) | P-value (crude) | P-value (adjusted) |
|---|---|---|---|---|---|---|
| | 557 | 315 (56.6) | | | | |
| Sex | | | | | | |
|    Female | 277 | 157 (56.7) | 1.0 (ref) | 1.0 (ref) | | |
|    Male | 280 | 158 (56.4) | 0.99 (0.71–1.38) | 0.97 (0.66–1.42) | 0.953 | 0.872 |
| | | | | | | |
| Age | | | | | | |
|    0–≤5 years | 135 | 72 (53.3) | 1.0 (ref) | 1.0 (ref) | | |
|    5–≤10 years | 246 | 182 (74.0) | 2.49 (1.6–3.88) | 2.45 (1.57–3.84) | | |
|    10–≤14 years | 78 | 46 (59.0) | 1.26 (0.72–2.22) | 1.27 (0.72–2.25) | | |
|    14–≤18 years | 21 | 10 (47.6) | 0.8 (0.31–2.01) | 0.8 (0.31–2.04) | | |
|    >18 years | 77 | 5 (6.5) | 0.06 (0.02–0.15) | 0.063 (0.02–0.15) | <0.001 | <0.001 |
| Housing | | | | | | |
|    No crowding[**] | 276 | 141 (51.1) | 1.0 (ref) | 1.0 (ref) | | |
|    Crowding | 281 | 174 (61.9) | 1.56 (1.11–2.18) | 1.1 (0.75–1.62) | 0.01 | 0.62 |
| Toilets | | | | | | |
|    Outside[***] | 256 | 157 (61.3) | 1.0 (ref) | 1.0 (ref) | | |
|    Inside | 301 | 158 (52.5) | 0.7 (0.5–0.98) | 0.75 (0.48–1.15) | 0.036 | 0.191 |
| Water | | | | | | |
|    Outside[****] | 100 | 57 (57.0) | 1.0 (ref) | 1.0 (ref) | | |
|    Inside | 457 | 258 (56.5) | 0.98 (0.63–1.51) | 1.41 (0.82–2.42) | 0.921 | 0.209 |

**Notes.**

N, patients enrolled with suspected viral hepatitis; n, patients with acute hepatitis A; OR, Odds Ratio; CI, confidence interval; ref, reference group.

[*]Adjusted for: sex, age, housing, toilets and water.

[**]CROWDING defined as >2.5 persons/room.

[***]TOILET outside = outside or latrine.

[****]WATER outside = water from a well.

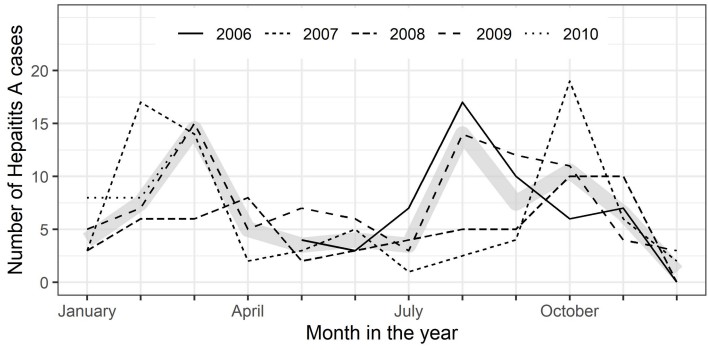

**Figure 2  City of León: seasonality of symptomatic hepatitis A cases 2006–2010.** Monthly numbers of enrolled symptomatic hepatitis A cases by years (2006 –2010); thick gray line represents median number of cases per month.

## DISCUSSION

*Study population*: In our study 80.6 of the 315 symptomatic hepatitis A cases enrolled were children aged ≤10 years, an age where HAV infection is in the majority asymptomatic (*Armstrong & Bell, 2002*). This implies that many more subclinical HAV infections must have occurred in parallel in the population surveyed. Overall, our findings indicate that Nicaragua was in 2006–2010 still a country of high endemicity and that the single-dose hepatitis A vaccination study (*Mayorga et al., 2016*)—for which the present project acted as a kind of community control group—had been conducted during a sustained circulation of HAV.

*Seroepidemiology*: Quantitative anti-HAV IgG antibody levels could only be measured in 263 of the 557 patients with suspected hepatitis. In the Non-A subpopulation tested ($n = 94$), the age-specific seroprotection rates rise until the age of 11- ≤16 years, where 100% seroprotection is reached (Table 2), consistent with the 100% found from age 18 years onwards in our 2003 study (*Mayorga Perez et al., 2014*). The present anti-HAV antibody data cannot be used as cross-sectional population seroprevalence data, as only symptomatic subjects were screened. However, the data gathered allow the conclusion that hepatitis A was still highly endemic in the area investigated: nine of the 11 Non-A patients lacking seroprotection were ≤11 years of age and only two were 18 and 19 years of age. Nevertheless, the incomplete seroprotection of 71.4% observed in the age group 16 to ≤20 years, and the fact that two of the five Non-A infants were seronegative, i.e., their mothers most likely as well, indicates the beginning of a transition away from high endemicity. The most recent hepatitis A data cover the years 2004–2012 and report for the whole country a rise from 19.2 to 36.9 infections per 100,000 inhabitants, and for León a rise from 12.6 to 22.2 (2011), followed by a sharp increase to 86.9/100.000 in 2012 (Supplemental File 4). Our monthly enrolment of an average 6.7 patients with acute hepatitis A matches well with the average of 6.3 cases reported monthly for León during 2006–2010 (Supplemental File 4). Using our 2003 cross-sectional serosurvey data (*Mayorga Perez et al., 2014*), a recent literature review on hepatitis A epidemiology in Latin American countries classified the endemicity level of Nicaragua as high-intermediate (*Andani et al., 2020*), based on the age at midpoint of population immunity (*Mohd Hanafiah, Jacobsen & Wiersma, 2011*) of 5–9 years. Previous studies in the same area in León showed that there was already between 1995–96 and 2003 a halving of the HAV infection risk for 1 to 5 years old children (*Mayorga Perez et al., 2014*). Although we found with the limited serosurvey data only few serological indications of shifting endemicity among the non-A cases, i.e., very few HAV susceptible school children or young adults (Table 2), 17.8% of the acute hepatitis A cases occurred between 10 and ≤18 years ($n = 56$) and 1.6% ($n = 5$) at >18 years of age. In addition, there were two cases in infants who are in hyperendemic settings usually seroprotected by maternal antibodies (*Brinkhof et al., 2013*). Taken all together, these signs of a beginning epidemiological shift are a call for the health authorities to start monitoring hepatitis A more closely, e.g., with repeated cross-sectional serosurveys (*Cutts & Hanson, 2016*; *Mayorga Perez et al., 2014*), in order to not miss the right time to start with public health interventions, such as e.g., universal childhood vaccination.

*Clinical features*: The clinical outcome of HAV infection is strongly associated with age: below 6 years of age only 10–20% of infections are symptomatic, at the age of 10 years up to 50% might present jaundice or other clinical features and from age 18-20 years upwards 80–90% of infections are symptomatic (*Armstrong & Bell, 2002*). The severity of disease and fatal outcomes are higher in older age groups, with case fatality rising from 0.1% in children <15 years, to 0.3% among persons 15–39 years of age and reaching 1.8–5.4% above 50 years of age (*Lemon et al., 2018*). In our study, no severe cases were seen, the laboratory findings were 'unspectacular' and corresponded to the overall mild clinical features of the patients enrolled. None of the children or adolescents had to be hospitalized, which is typical for a country where hepatitis A is highly endemic and most infections occur during childhood (*World Health Organization, 2010a*). However, there are reports from Latin American countries of severe cases with fulminant hepatitis A and acute liver failure (*Ciocca et al., 2007*; *Ferreira et al., 2008*), this being an important reason for implementation of UMV of children (*Lemon et al., 2018*), as e.g., in Argentina (*Vizzotti et al., 2014*). It is still debated whether differences between various geographic world regions regarding pathogenicity and virulence of HAV are due to genetic host factors in certain ethnic groups or due to nucleotide sequence variations in certain HAV subtypes (*Long et al., 2014*; *Vaughan et al., 2014*).

*Socioeconomic factors*: All over the world the risk of HAV infection is highly correlated with the socioeconomic status (*Jacobsen & Koopman, 2005*). Although the global number of HAV infections is still on the rise (*World Health Organisation, 2012)*), declining risks of infections are seen in many countries over the last two decades, mainly due to rising sanitary and economic standards and better access to clean water (*Jacobsen & Koopman, 2005*), but in some countries also due to UMV, such as in Israel (*Levine et al., 2015*), Argentina (*Vizzotti et al., 2014*) and Brazil (*Souto, de Brito & Fontes, 2019*). This shift from high to intermediate HAV endemicity has been observed in Latin America since the 1990s, a trend which continued in the 2000s (*Jacobsen & Wiersma, 2010*; *World Health Organization 2010b*; *Andani et al., 2020*). The risk for HAV infection is influenced by many different factors. Logistic regression analysis showed in Brazil that anti-HAV seroprevalence correlates with increasing age and level of crowding (*Vitral et al., 2014*), whereas access to clean water and reinforced concrete buildings were protective factors (*Vitral et al., 2012*). For age and crowding we can see similar effects in our study, with age being the most strongly correlated factor. For the water source we saw no significant effect, even though this is mentioned in many studies from Latin America (*Jacobsen & Wiersma, 2010*), probably due to fact that the water was clean in our study area, even if it came from an outside well. In a Brazilian study, an overall low socioeconomic status was identified as risk factor (*Vitral et al., 2012*). We could not assess this in our study, because we did not evaluate income and education of the families. Thus, the socioeconomic risk factors for HAV identified in our study were only the sanitary conditions, i.e., the place of the toilet, and crowding. Household contacts are an important source of infection (*Lima et al., 2014*). In our setting, crowding made a significant difference in crude Odds ratios but not in the adjusted ones. Also, we could not see a difference between households hosting different age groups. Another Brazilian study suggests that crowding is confounded by other socioeconomic

factors, possibly due to the difficulty in maintaining hygiene in big households (*Almeida et al., 2001*). In our study this is shown with the adjusted odds ratio, where the *p*-value for crowding is no longer significant, indicating that there is a confounding by socioeconomic factors, such as the localization of toilets (Table 3). In the logistic regression analysis we saw a correlation between the different risk factors. Independently seen, none of the factors made, however, a significant difference. Therefore, the main risk factor appears to be the low socioeconomic status with all his features.

*Seasonality*: In our study there were slightly more hepatitis A cases in the 1st and 4th quarter of the year, i.e., during autumn/winter, when the weather is in Nicaragua slightly cooler and drier. There are seemingly conflicting data published on hepatitis A seasonality, some reports indicating peak incidences in the wet season or summer, as for Brazil (*Villar, de Paula & Gaspar, 2002*), others in the dry season or winter, as in Israel (*Green et al., 2001*). Many different environmental factors, such as rainfall, ambient temperature, and air humidity, as well as host behavior, source of infection and mode of transmission are playing a role in the seasonality of infectious diseases (*Martinez, 2018*). A recent review on the seasonality of viral hepatitis explains these seemingly conflicting seasonality patterns for hepatitis A infections with the different modes of transmission being decisive: water-borne during wet summer seasons in case of low hygiene standards and rather food-born in winter, caused e.g., through shellfish consumption (*Fares, 2015*). Interestingly, even in present-day Spain water-related climate events correlate with the incidence of hepatitis A (*Gullón et al., 2017*).

Our study has **limitations**: The data collection was done via passive surveillance, relying on self-reporting of cases (parents/patients). The real number of acute hepatitis A cases was most likely higher, as we collected the more symptomatic cases, and because of the low manifestation index in young children, many more asymptomatic HAV infections must have occurred during the observation period. We collected educational levels and jobs only for the enrolled subjects themselves, and not for their mothers/parents, which would have made more sense because most of the patients were young children. Anti-HAV IgG antibody data were only available for 47% of all 557 cases of assumed viral hepatitis and in only 94 of the 242 Non-A cases, limiting the significance of the age-specific seroprotection data, although they pretty well match previous data from the same area (*Mayorga Perez et al., 2014*). The number of 315 hepatitis A cases was too small to assess in the given setting with certainty the influence of all socioeconomic factors on contracting acute hepatitis A.

The **strengths** of the study were: In a hepatitis A endemic setting, a large series of cases with assumed viral hepatitis was carefully ascertained and laboratory confirmed, allowing to accurately describe the clinical features of hepatitis A. Due to the prospective collection of viral hepatitis cases over all seasons of 4 years, the seasonality features of hepatitis A could be assessed and documented for the first time for Nicaragua. The screening test diagnosis of acute hepatitis A was confirmed by a recognized reference laboratory.

## CONCLUSIONS

Nicaragua is a country of still high hepatitis A endemicity, with all its typical features. The majority of cases were oligosymptomatic with no severe pathology seen in over three

hundred documented cases. The main risk factor for acquiring hepatitis A is living in low socioeconomic conditions. There are some first indications that the endemicity level started in the 2000s shifting from high to high-intermediate. A closer monitoring of hepatitis A is warranted in Nicaragua in order to be able to start in time with appropriate public health interventions.

## ACKNOWLEDGEMENTS

Many thanks go to the staff of the health centers in León, and to the study nurses and laboratory technicians of the Department of Microbiology and Parasitology of the university UNAN in León.

### Funding
The project was supported by an unrestricted research grant by Crucell Switzerland AG (formerly Berna Biotech), Bern, Switzerland. The funders had no role in study design, data collection and analysis, decision to publish, or preparation of the manuscript.

### Grant Disclosures
The following grant information was disclosed by the authors:
Crucell Switzerland AG (formerly Berna Biotech), Bern, Switzerland.

### Competing Interests
The authors declare there are no competing interests.

### Author Contributions
- Sophie Jaisli, Marcel Zwahlen and Sereina A. Herzog analyzed the data, prepared figures and/or tables, authored or reviewed drafts of the paper, and approved the final draft.
- Orlando Mayorga conceived and designed the experiments, performed the experiments, authored or reviewed drafts of the paper, and approved the final draft.
- Nadia Flores conceived and designed the experiments, performed the experiments, authored or reviewed drafts of the paper, monitored and supervised the health centers, and approved the final draft.
- Sandra de Berti performed the experiments, authored or reviewed drafts of the paper, was responsible for study logistics, and approved the final draft.
- Gustav Frösner performed the experiments, authored or reviewed drafts of the paper, performed the IgM and IgG Hepatitis A antibody tests, and approved the final draft.
- Christian Herzog conceived and designed the experiments, analyzed the data, authored or reviewed drafts of the paper, and approved the final draft.

### Human Ethics
The following information was supplied relating to ethical approvals (i.e., approving body and any reference numbers):

The project was approved by the Medical Faculty of the National Anonymous University of Nicaragua (UNAN), León.

## Data Availability

The raw data are available in the Supplemental Files.

## Supplemental Information

Supplemental information for this article can be found online at http://dx.doi.org/10.7717/peerj.11516#supplemental-information.

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
