# Peer review of "Clinical, serological and epidemiological features of hepatitis A in León, Nicaragua"

_PeerJ, doi:10.7717/peerj.11516_

## Round 0.1 · original submission · Major Revisions

The review process is now complete, and two thorough reviews from highly qualified referees are included at the bottom of this letter. All reviewers including myself agree the manuscript deserves to be published. Although there is considerable merit in your paper, we also identified some concerns that must be considered in your resubmission. Please, provide a study design flow chart and the exclusion and inclusion criteria used. Please, also provide and improve tables to allow inferences from the readers.

Reviewer 1 ·

Basic reporting

The manuscript clearly described prevalence, clinical features and risk factors of acute hepatitis A in León, Nicaragua during 2006-2010.

Experimental design

In this study cases enrolled are symtomatic cases presented to healthcare facility by advertise, some of asymtomatic/mild cases might not be included.

Validity of the findings

This study reveal the informative epidemiologic data of acute hepatitis A in León, Nicaragua during 2006-2010.

Additional comments

Reviewer's report: The manuscript entitled "Clinical, serological and epidemiological features of hepatitis A in León, Nicaragua
” This study described prevalence, clinical features and risk factors of acute
hepatitis A in León, Nicaragua during 2006-2010. The majority of cases are young population. In the basic knowledge, HAV infection is mostly asymtomatic among young children, In this study cases enrolled are symtomatic cases presented to healthcare facility some of asymtomatic/mild cases might not be included. This study reveal the informative epidemiologic data of acute hepatitis A in León, Nicaragua during 2006-2010.

Reviewer comments
1.Result part: Authors should add table of demographic data/clinical data of enrolled cases, classified by age group, this would be informative table. And result in clinical part and biochemical lab should be summarized in the table.

Quality of written English: Acceptable

Declaration of competing interests: I declare that I have no competing interests.

Reviewer 2 ·

Basic reporting

The subject of this study is interesting for the high endemicity of hepatitis A infection, however, the presentation of the data needs to be revised to make its readability.

Experimental design

The study design could provide a flow chart to make it clear.

Validity of the findings

Figure 1 is published, this could be confusing its originality.
The tables are suggested to revise, then clearly presented in the main text.
Finally, the presentation of the validity of the findings should be supported by measurements.

Additional comments

1. In the abstract, the " the aim of the study " please clarify. Is the statement of the aim of the study consistent with the title or main text?
2. In the study conduct, it is suggested to add a time flow chart of baseline survey and intervention.
3. Please provide the inclusion and exclusion criteria clearly.
4. Were the participants excluded including vaccination or past infection?
5. In the Results, figure 1, Does it present in the previous study? (Mayorga et al., 2016)Did it provide new information in this study?
6. In the Results,
As we can see from the raw data, it is suggested to revise table 2 or the main text. Given the total number of participants and the number, measurements are inconsistent with Table 2, the readers hardly understand the result in the main text and Table 2.
As we can see in Table 2, the trend of seroprevalence is increasing with age, even the small sample size in each aged group. It may be compared between the measurement group and the symptomatic group.
7. In the Results,
As we can see from the raw data, it is suggested to revise table 1, given it provides very little information.
It may be compared between the measurement group and symptomatic group including clinical features and laboratory findings.
8. In the Results, ‘’ Logistic regression analyses”, it is suggested to exclude missing data (symptomatic group), even past infection cases, or subclinical cases. This could be more exact to predict the risk factors.
9. In the Discusses, the authors emphasized the high endemicity in this area, it is suggested to explain by using the data of measurement group or by aging.

Annotated reviews are not available for download in order to protect the identity of reviewers who chose to remain anonymous.

---

## Round 0.2 · accepted · Accept

The authors have satisfactorily responded to all questions and made the necessary changes to the manuscript.

Reviewer 2 ·

Basic reporting

no comment

Experimental design

no comment

Validity of the findings

no comment

Additional comments

The article entitled " Clinical, serological and epidemiological features of hepatitis A in León, Nicaragua’’ has been reviewed again. After the revision, the manuscript could be present as the clinical, serological, and epidemiological features of hepatitis A.
However, the author’s response to query No.1. in the “abstract”, the " the aim of the study " is not described more clearly. “To monitor and document endemicity and disease burden of acute hepatitis A” mean “The clinical manifestations and seroprevalence of HAV” have been reported in the previous study or in this study. The “an ongoing vaccine effectiveness study” is ongoing in the future or now in this study. Most studies may have one objective, the introduction reported three objectives. I felt confused. The statement of the aim of the study in the abstract and introduction was suggested to compatible with the title.
The study flow chart, presentation of the data, and the tables have been revised.